# A QSAR Study for Antileishmanial 2-Phenyl-2,3-dihydrobenzofurans [note 1]

**DOI:** 10.3390/molecules28083399

**Published:** 2023-04-12

**Authors:** Freddy A. Bernal, Thomas J. Schmidt

**Affiliations:** University of Münster, Institute of Pharmaceutical Biology and Phytochemistry (IPBP), PharmaCampus—Corrensstraße 48, 48149 Münster, Germany; freddy.bernal@leibniz-hki.de

**Keywords:** 2-phenyl-2,3-dihydrobenzofurans, *Leishmania*, 3D-QSAR, QSAR, neolignan analogues

## Abstract

Leishmaniasis, a parasitic disease that represents a threat to the life of millions of people around the globe, is currently lacking effective treatments. We have previously reported on the antileishmanial activity of a series of synthetic 2-phenyl-2,3-dihydrobenzofurans and some qualitative structure–activity relationships within this set of neolignan analogues. Therefore, in the present study, various quantitative structure–activity relationship (QSAR) models were created to explain and predict the antileishmanial activity of these compounds. Comparing the performance of QSAR models based on molecular descriptors and multiple linear regression, random forest, and support vector regression with models based on 3D molecular structures and their interaction fields (MIFs) with partial least squares regression, it turned out that the latter (i.e., 3D-QSAR models) were clearly superior to the former. MIF analysis for the best-performing and statistically most robust 3D-QSAR model revealed the most important structural features required for antileishmanial activity. Thus, this model can guide decision-making during further development by predicting the activity of potentially new leishmanicidal dihydrobenzofurans before synthesis.

## 1. Introduction

The World Health Organization (WHO) has recognized Leishmaniasis as a public health concern, being one of the so-called Neglected Tropical Diseases (NTDs). It has been estimated that 600,000 to 1 million people are infected every year with the various forms of Leishmaniasis, primarily in tropical and subtropical regions [1,2], and despite control and surveillance campaigns, the panorama has worsened lately with clear outbreaks due to management issues associated with the COVID-19 pandemic [1]. Even though the disease burden (in disability-adjusted life years, DALYs) had reduced by 5.4% from 2015 to 2019 [1], it is still considerably high (>600,000 DALYs [3]) with more than 230,000 newly reported cases in 2021 [1]. Caused by parasites of the genus *Leishmania*, it exists in several clinical forms related to the particular species affecting the host [2,4,5]. Current treatments are inadequate, displaying several drawbacks including, but not limited to, high toxicity and poor efficacy [3,6,7]. Different institutions and research groups have put great effort into the search for antileishmanials [3,8,9,10,11,12], however, effective drugs remain to be found, especially for Visceral Leishmaniasis, the most aggressive form of the disease.

The use of computational methods to aid in solving different problems in drug discovery pipelines is becoming more and more important, particularly with the advent of artificial intelligence [13,14,15,16,17,18,19]. In silico approaches have, therefore, been applied for the rational design and discovery of potential drugs against Leishmaniasis [20,21,22,23,24,25]. Due to the limited knowledge about validated targets for Leishmaniasis, ligand-based methods investigating structure–activity relationships (SARs) may represent a suitable approach. During our research program for fighting Leishmaniasis using natural products and natural product-like small molecules, we have reported a series of 2-phenyl-2,3-dihydrobenzofurans, synthetic analogues of natural dihydrobenzofuran neolignans, with antileishmanial potential [26]. In such a study, from qualitative inspection of the compounds’ structure and activity, it became evident that an in-depth study focusing on quantitative structure–activity relationships (QSAR) for this series of compounds would be interesting. Therefore, we present herein a comparative QSAR study for antileishmanial 2-phenyl-2,3-dihydrobenzofurans, using different machine learning methods and molecular descriptors, as well as 3D-QSAR. The various models’ statistical performance was assessed exhaustively using a comprehensive set of existing quality metrics and compared between the different approaches. Key structural features conferring activity were finally deduced from the best-performing model.

## 2. Results and Discussion

### 2.1. Data Set

The data set used in the present study comprises a series of seventy congeneric *trans*-2-phenyl-2,3-dihydrobenzofurans with antileishmanial potential previously synthesized and reported by us [26]. According to their structural features, two different groups, **A** and **B** can be easily distinguished (Figure 1). The full list of the individual compounds’ structures used in this study and their antileishmanial activity are reported in Appendix A. Compounds in class **A** have natural product-like structures fairly close to neolignans commonly found in plants [27,28], which are biosynthesized as dimers of phenylpropenoid building blocks. Compounds of class **B** are synthetic analogues lacking the characteristic dimeric nature of natural products. The selected compounds possess widely distributed antileishmanial activity, with IC_50_ values against axenic amastigotes of *Leishmania donovani* ranging from 0.5 to >200 μM (i.e., covering almost three orders of magnitude), which makes them amenable for QSAR analyses.

### 2.2. QSAR Modeling

3D molecular models for each compound (Appendix A) were obtained by energy minimization of the lowest energy conformer from a conformational search, using the semi-empirical AM1 method. The resulting structures were used to calculate molecular descriptors for the purpose of machine learning (ML)-based QSAR and aligned for 3D-QSAR modeling. Using the Molecular Operating Environment (MOE) software [29], a total set of 435 molecular descriptors was obtained. Feature selection through contingency analysis, as implemented in MOE (Appendix A), led to a reduced set of 107 descriptors, corresponding to those with the highest utility for QSAR modeling (see Section 3.1 for details; for the full list of descriptors see Appendix A). Multiple Linear Regression (MLR), Random Forest (RF), and Support Vector Regression (SVR) were used as learning algorithms for the training of descriptor-based models. On the other hand, the structures prepared as mentioned above were aligned using Open3DAlign [30], whereupon Open3DQSAR [31] was employed to train 3D-QSAR models using Partial Least Squares (PLS) regression. Details for each model type are presented below. Three different combinations of training/test sets were used in each case by the random splitting of the samples (percentage ratio of 74/26 unless otherwise stated).

#### 2.2.1. Model Building

As a first approach, MLR was used to train models MLR1-MLR3. In all cases, a maximum number of 5 to 6 features were employed, thus being within the fifth of the samples in the training set, as recommended by the Organization for Economic Co-operation and Development (OECD) [32]. A genetic algorithm (GA) [32,33,34] was then used for efficient feature selection. Several independent searches with a different fixed number of features (3, 4, 5, and 6 descriptors) were performed. From each run, the top five models according to $Q^{2}$ (coefficient of determination during leave-one-out cross-validation, CV) were kept for further analysis (mathematical equations representing models MLR1-MLR3 are shown in Appendix A).

During our second descriptor-based approach, models were independently trained using RF and SVR (models RF1–RF3 and SVR1–SVR3, respectively). These algorithms, unlike MLR, are not limited to strict linear correlation and might hence perform better in the case of nonlinear SAR. Hyperparameter optimization was independently performed in each case to obtain the best possible outcome (according to CV).

Finally, a CoMFA-like method was applied to generate models 3D1–3D3, based on Molecular Interaction Field (MIF) calculations and PLS. The three models differed in the composition of training/test sets at a constant splitting ratio. MIFs were obtained using the MMFF94 force field, as implemented in the Open3DQSAR [31] software. Feature selection (see experimental section) led to data matrices typically exceeding 2000 variables. The optimum number of final latent variables (LV) for PLS was chosen by CV.

#### 2.2.2. Model Validation

It is well-known that a high $Q^{2}$ on its own does not assure good predictive power [35]. Nor necessarily does a high $R_{pred}^{2}$, mainly due to its strong dependence on the selection of the training set [36]. Therefore, the performance of the generated models was validated through exhaustive statistical assessment (Appendix A) using thirteen different metrics commonly accepted within the QSAR community (Appendix A). The use of such statistical parameters ensured a comprehensive assessment of model performance; however, this was accompanied by a practical limitation in terms of model comparison. Thus, dimensionality reduction by Principal Component Analysis (PCA) offered a simple solution for a qualitative comparison among models (Appendix A). It was then evident that SVR and MLR models performed similarly.

The relative variability among validation metrics was seemingly low as evidenced by the corresponding coefficient of variation, demonstrating that, for most of the statistical parameters, all the models performed rather comparably (Appendix A). In addition to directly comparing all the metrics for the validated models, two consensus scores (*F*1 and *F*2) were calculated (Figure 2). We have already shown the utility of such a strategy for validation and model performance assessment in regression problems [37] (see Appendix A, for score definitions). The *F*1 score denotes the number of statistical parameters within typical or commonly established thresholds (i.e., the number of positive assessments). *F*2 assigns either a reward or a penalization for each statistical parameter included to reflect compliance with established thresholds in order to assure good performance (the higher *F*2, the better the model) [37]. Since it was observed that *F*1 alone might lead to misinterpretations for models with poor CV statistics, *F*2 was exclusively calculated for models with $Q^{2}$ and $R_{pred}^{2}$ above 0.5, thus guaranteeing more stringent criteria.

Evidently, important differences among models became obvious through the use of both consensus scores. According to *F*1, the 3D-QSAR models (3D1–3D3) clearly outperformed all the other models, without failing any criteria (i.e., complying with all the parameters’ thresholds used herein). Models RF1 and RF2 showed compliance with 10 out of 12 criteria, however, the latter model might suffer from overfitting as suspected from poor CV statistics ($Q^{2}$ < 0.5). Models SVR1 and SVR2 showed compliance with 8 validation criteria, and although MLR1-MLR3 exhibited the same level of compliance, SVR models afforded generally better *F*2 scores. Thus, even the best MLR model (MLR1) performed worse than the best SVR and RF models (SVR1 and RF1, respectively, Figure 2). According to both consensus scores, *F*1 and *F*2, the MIF-based 3D-QSAR is the method of choice for modeling the data set under study.

The effect of training/test size on model performance was analyzed by changing the respective ratio from 74:26 to 70:30 (model 3D4) and 80:20 (model 3D5). As can be seen, the newly generated models displayed higher *F*2 values (Figure 2) than 3D1–3D3. Thus, regardless of the training set size, 3D-QSAR models are comparably good predictors.

Owing to their higher *F*2 scores, the quality of models 3D4 and 3D5 was further investigated through the determination of their robustness. The progressive scrambling method [38], as implemented in Open3DQSAR, was used to achieve this ultimate comparison. The method calculates a normalized correlation coefficient (${Q_{0}^{*}}^{2}$) resulting from the fitting of $Q^{2}$ and $R^{2}$ after progressive scrambling, which can be interpreted in the same manner as a normal $Q^{2}$ value [38] (i.e., the higher the better). The calculated ${Q_{0}^{*}}^{2}$ for models 3D4 and 3D5 was 0.61 and 0.59, respectively. The rather subtle difference would suggest that both models are equally robust. Nevertheless, subsequent analysis was performed with 3D4 as the nominally best model.

#### 2.2.3. Applicability Domain for Model 3D4

Once having a valid and robust model in hands, determination of the applicability domain (AD) was mandatory in order to fulfill another OECD requirement [32]. The AD in its currently accepted definition is the response and chemical structure space in which the model makes predictions with a given reliability [39]. Therefore, it constitutes a fundamental point to assure the correct use of any model when the prediction of new, unseen compounds is desired. Within the plethora of existing methods for defining AD [39,40], the leverage method [39,40,41,42] (a distance-based method) was used in the present research. The corresponding Williams plot (standardized residuals vs. leverage) is shown in Figure 3A. It becomes obvious that none of the compounds in the test set appeared beyond the “warning leverage” (denoted *h** and represented by the vertical dashed line), indicating that all of them are within the AD of the model. Leverage values higher than *h** in the test set would have indicated unreliable predictions as a result of substantial extrapolation [39,41]. Notably, compounds **24** and **21** (see Appendix A, for structures), members of the training set, have leverages higher than *h**, showing their significant influence on the regression model. Both of them were accurately predicted (low standardized residuals). On the other hand, compound **13** (see Figure 3B) yielded a relatively large standardized residual, demonstrating that its activity was not entirely well-predicted by the model, although still located within an accepted range (<2.5σ). Compliance with AD during model building and validation, as demonstrated above for all the compounds used, represents yet another strength of model 3D4. The combination of good validation statistics with proven robustness and well-defined AD, therefore, makes it a reliable model for the prediction of the antileishmanial activity of dihydrobenzofurans. Activity predictions by model 3D4 are summarized in Appendix A.

#### 2.2.4. Model Interpretation

One of the most important goals of QSAR models, in addition to predicting the activity of new compounds, is their interpretation in order to rationalize the underlying SARs [43]. Interpretation is particularly straightforward in the case of CoMFA and other 3D-QSAR variants based on MIFs, due to the implicit easiness of visualization [44]. Therefore, an analysis of the key structural features affecting the antileishmanial activity was carried out by inspecting the MIFs-derived contour maps for model 3D4. Such maps represent MIF regions with a high impact on the PLS regression model and are generated by plotting the PLS coefficients of MIF regions with absolute values higher than a certain threshold. CoMFA maps for 3D4 are shown in Figure 3C,D around the structures of compounds **13** and **30**, as representative potent and inactive compounds, respectively (see Figure 3B for chemical structures). Van der Waals interactions (Figure 3C) around the substituent on position C-5 showed a positive effect on the activity, being probably the most important characteristic (green contours V_1_ and V_2_). This was in very good agreement with previously observed qualitative SARs [26]. Compounds bearing an acrylate unit were generally more active than those without it, which, according to the 3D-QSAR model, is partially due to increased steric bulk in that region. Moreover, compounds with bulky alkoxy groups obtained by esterification of the acrylate moiety were more active (contour V_2_). However, the steric MIFs also indicated a small negative effect on activity in cases where this substituent was too large (as found for compound **16**, yellow contour V_3_). The presence of substituents on positions C-3′ and C-5′ (pending phenyl ring) were determined as positively affecting the antileishmanial activity, too (contour V_4_). A similar trend was evident for the substituents on the carboxy group attached to C-3 (green contour V_5_). On the other hand, analysis of the electrostatic interaction field (Figure 3D) revealed that the presence of electron-rich chains on C-5 increased the activity (big red contour E_1_). In contrast, some electron deficiency on the aromatic ring near C-5 might improve the activity (blue contour E_2_). Electron deficiency on the pending phenyl group resulted in a favorable effect on activity (blue contour E_3_) as well, whereas H-bond donors on C-3′ and C-5′ led to superior activity (red contours E_4_). The MIFs also indicated that the establishment of H-bonds by the carboxy group on C-3 might improve the activity (both donor and acceptor nature; contours E_5_).

The steric field contributed to a larger extent to the explanation of the variance in activity (62.43% steric vs. 37.57% electrostatic) in model 3D4, suggesting that increasing the lateral chain sizes to a certain optimum played a more important role than, for instance, changing electron density on the benzofuran moiety. All the observations and conclusions retrieved from this model were in full agreement with the reported qualitative SAR analyses [26].

## 3. Materials and Methods

### 3.1. Data Preparation

A basic preparation of the data set was carried out for all the compounds included as follows: 2D structures of the *trans*-2-phenyl-2,3-dihydrobenzofurans were converted into 3D models assuring a (2*R*)-configuration in the Molecular Operating Environment (MOE) software (version 2018.0101) [29]. Since all compounds were synthesized and tested as racemates, this does not imply that the *R*-enantiomers are the eutomers. Each structure was then submitted to energy minimization using the Amber10:EHT molecular field. Subsequently, a conformational search using the LowMD mode in MOE within an energy window of 5 kcal/mol and RMSD limit of 0.75 Å was performed. The structures of the lowest energy conformers were refined by energy minimization using the semi-empirical AM1 method with the MOPAC module of MOE. The obtained 3D structures were used for the calculation of the whole set of 435 molecular descriptors available in MOE. The suitability of the molecular descriptors for QSAR purposes was assessed by contingency analysis as implemented in MOE. Minimum threshold values of 0.6 for the contingency coefficient and 0.2 for Cramér’s V, uncertainty, and correlation coefficients were chosen for the selection of 107 descriptors to be used in the QSAR study (Appendix A, for the final list of descriptors used). Activity data (Appendix A) were used in the form of the negative decadic logarithm (pIC_50_) of the half-maximal inhibitory concentration (IC_50_ in mol/L).

### 3.2. Multiple Linear Regression Models

The data set was divided into training and test sets as follows: the compounds were sorted in descending order of activity (pIC_50_) and 18 different bins were defined. From each bin, a compound was randomly selected and assigned to a test set representing 26% of the samples. The process was repeated several times to obtain different training/test set compositions for model building.

QSAR models were then built for those data sets using the genetic algorithm-driven variable selection and multiple linear regression analysis (GA/MLR) [32,33,34]. The GA/MLR algorithm was obtained from the CCG/MOE SVL exchange website (script GA.svl) [45]. A fixed number of variables was used in all cases and models with 3 to 7 variables generated for each training set. For each GA/MLR run, a set of 100 models was generated. Each GA run had a maximum of 1000 evolution cycles as termination criteria. In each case, the five models with the highest $Q^{2}$ during leave-one-out (LOO) cross-validation (CV) were tested for external predictivity by calculating the activity of the test set compounds.

### 3.3. Random Forest Models

Random forest (RF) is an ensemble learning method based on the use of a group of decision trees [46,47]. A bootstrapped sample of data is employed for the training of each tree, typically considering a randomly selected subset of features during node splitting. The final predicted property is an average of all the predictions obtained from the individual decision trees. Reduction in the Gini index or Gini “impurity” drives node splitting [48]. Data sets prepared as described before were submitted to RF regression using Scikit-learn [49]. The number of trees in the forest, the minimum number of samples required to be at a leaf node, the minimum number of samples required to split an internal node, the maximum number of features to consider for the best split, and the number of samples to draw from the training set during bootstrap were subject of optimization in this work. A coarse-to-fine approach was followed to accomplish such a hyperparameter tuning. During the first instance, random sampling within the selected hyperparameter space was performed, while the second instance consisted of an exhaustive grid search. $Q^{2}$ statistics (5-fold CV) guided the selection of the best combination of hyperparameters. The corresponding Scikit-learn implementations were applied to successfully accomplish this sequential process.

### 3.4. Support Vector Machines Models

Support Vector Machines (SVM) attempt to segregate the data set into different classes of objects defining so-called hyperplanes [50]. Data points located close to the hyperplanes are denominated support vectors. Minimization of the gap between the support vectors delimiting a hyperplane (margin) drives the selection of the best hyperplanes. Kernel functions are typically required to help find the hyperplanes through a process of transformation from a lower to a higher dimensional space (i.e., increasing the dimensionality). Datasets prepared as described before were submitted to SVM regression using Scikit-learn [49]. The kernel function, the kernel coefficient gamma, the epsilon-tube (if applicable), and the regularization parameter C [50] were subject to optimization, using the same protocol as described for RF.

### 3.5. 3D-QSAR Models

The 3D molecular structures obtained as described above were aligned using the automatic alignment algorithm implemented in Open3DAlign [30]. Molecular interaction fields (MIFs) were calculated using the MMFF94 force field with default probes (neutral C atom in alkyl chain with sp^3^ hybridization for the steric MIF and a single positive punctual charge for the electrostatic MIF) with a 1.0 Å grid-step in a box of 28 × 30 × 23 Å (box big enough as to have a 5 Å out gap around the largest molecule). The number of obtained variables was reduced according to conventional cutoff limits (±30 kcal/mol). The remaining variables were scaled by the block unscaled weighting [51] algorithm implemented in Open3DQSAR [31] (version 2.281). Then, a variable selection procedure comprising Smart Region Definition [52] and Fractional Factorial Design [53] was carried out. Both algorithms were directly applied in Open3DQSAR. Finally, models were generated by Partial Least Squares (PLS) regression [54,55], using LOO-CV. The PLS coefficients were exported to be visualized in MOE as MIF contours. Different training and test sets were created by random splitting, where the latter constituted 20, 26, and 30% of the samples.

### 3.6. Statistical Validation

Quality assurance was assessed by calculating several statistical parameters [56], in addition to the conventional $R^{2}$ and $Q^{2}$, and are denoted as follows: $R_{0}^{2}$, ${R_{0}^{\prime }}^{2}$, *k*, and *k*′ [35], $R_{m}^{2}$ [36,57], mean absolute error (MAE) [58], $R_{pred}^{2}$ (=$Q_{F1}^{2}$) [59], $Q_{F2}^{2}$ [60], $Q_{F3}^{2}$ [61], and the concordance correlation coefficient (CCC) [62]. Definitions for those validation parameters are included in the Supporting Information (Appendix A). An in-house MATLAB script was employed for the simultaneous calculation of all the parameters. Thereafter, the models were scored using the global scoring functions *F*1 and *F*2, as previously reported [37] (see Appendix A for definition).

A comparison of the robustness of the two models with the best *F*2 scores was achieved using the statistical variation in the progressive scrambling method [38] as implemented in Open3DQSAR. Assessment of the applicability domain was carried out by the leverage approach [42,63], using an in-house MATLAB algorithm. The results were displayed as the corresponding Williams plot [42,63].

## 4. Conclusions

The promising antileishmanial potential of some 2-phenyl-2,3-dihydrobenzofurans, together with its evident structural dependence, encouraged us to thoroughly explore the structure–activity relationships underlying an existing medium size data set.

To this end, a considerable number of different QSAR models for the antileishmanial activity of the studied compounds were created. Three different machine learning methods trained on a matrix of 107 molecular descriptors, i.e., MLR, RF, and SVR, as well as 3D-QSAR based on the compounds’ MIFs were used to generate regression models. A comprehensive quality assessment by various validation metrics clearly demonstrated that 3D-QSAR models exhibited the best statistical quality, outperforming all descriptor-based models obtained with the other approaches. After evaluation of statistical robustness, model 3D4 was chosen to analyze the underlying MIFs for structural information quantitatively related to the antileishmanial activity. The significant role of an acrylate unit on C-5 was disclosed. Furthermore, a positive steric effect on the activity by bulky ester groups on that acrylate was confirmed. Substitutions on C-3′, C-4′, or C-5′ causing electron deficiency on the 2-phenyl ring might increase the activity, while H-bond donors on C-3′ and C-5′ would also improve it. Finally, the assessment of the applicability domain of the model emphasized the proper inclusion of all the studied compounds.

In summary, a complete statistical analysis and comparison of various QSAR models led to an exhaustively validated and robust final model able to predict the antileishmanial potency of 2-phenyl-2,3-dihydrobenzofurans. The major outcome of this research can thus be considered as a fundamental first-line tool for further analysis and development of this kind of compound to fight Leishmaniasis.

## Figures and Tables

**Figure 1 molecules-28-03399-f001:**
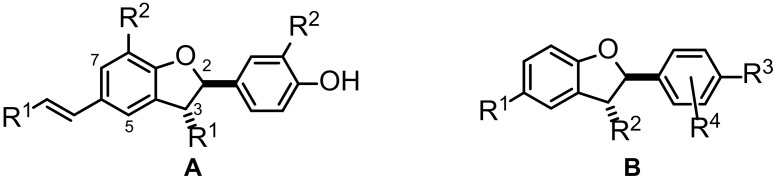
General structures of *trans*-2-phenyl-2,3-dihydrobenzofurans investigated in the present study. For all structures of the set of compounds under study, see Appendix A.

**Figure 2 molecules-28-03399-f002:**
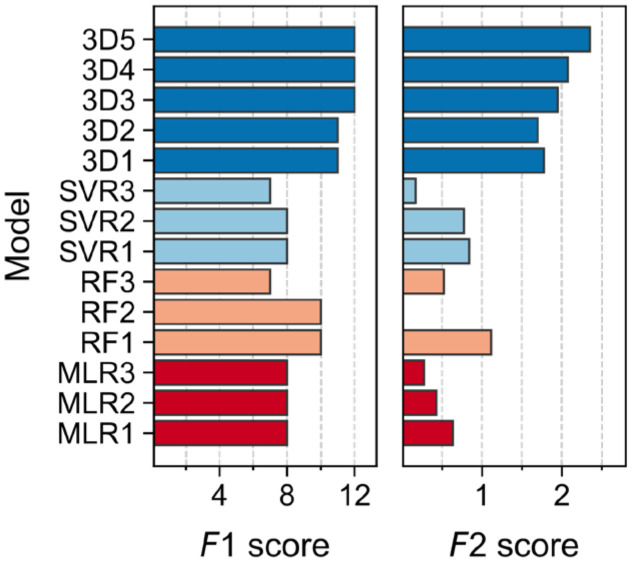
Consensus scores *F*1 and *F*2 for all the generated models according to [37]. Bars are colored by model type.

**Figure 3 molecules-28-03399-f003:**
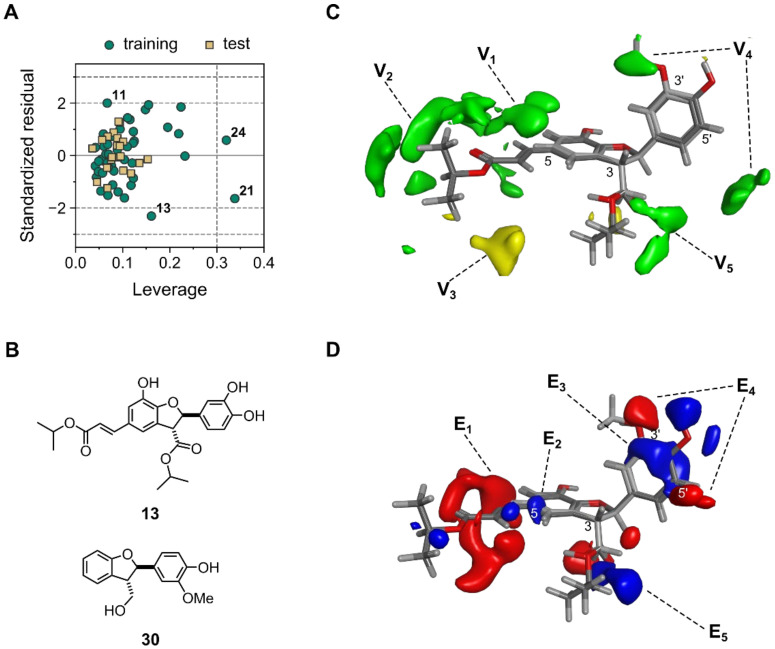
Applicability domain and interpretation of 3D-QSAR model 3D4. (**A**) Williams plot for AD definition of the model. Horizontal dashed lines represent 2σ and 3σ. The Vertical dashed line represents *h** (see text for interpretation). (**B**) Chemical structures of representative potent (compound **13**) and non-potent (compound **30**) antileishmanials. (**C**,**D**) MIF regions showing steric interactions affecting positively (green) and negatively (yellow) the activity (**C**), and electrostatic interactions by positively (blue) and negatively (red) charged regions positively affecting the activity (**D**). MIFs with a strong impact on activity according to model 3D4 (LV = 5) are plotted around the structure of compounds **13** (dark gray) and **30** (light gray).

## Data Availability

Data available from authors on reasonable request.

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
