# Peer review of "A QSAR Study for Antileishmanial 2-Phenyl-2,3-dihydrobenzofurans †"

_molecules, 2023, doi:10.3390/molecules28083399_

Round 1

Reviewer 1 Report

The manuscript "A QSAR Study for Antileishmanial 2-Phenyl-2,3-Dihydroben-2 zofurans" describes the use of computational models in development of drug design. The authors used the QSAR modelling for building a model to target neglected tropical disease Leishmaniasis. The work is compiled orderly and represented in well-defined manner in the manuscript. The few minor points to consider while final publication are given below.

1. The abstract is too descriptive. It should be more results oriented. 

2. The authors are advised to add the statistics of disease in the introduction to understand the mortality and severity of the disease. 

3. In manuscript, the final model optimized should be described with parameters obtained from the studies. 

4. The results obtained need to be explained in details with more graphics obtained while study.

5. The manuscript need to be revised for small grammatical mistakes. 

Author Response

The manuscript "A QSAR Study for Antileishmanial 2-Phenyl-2,3-Dihydroben-2 zofurans" describes the use of computational models in development of drug design. The authors used the QSAR modelling for building a model to target neglected tropical disease Leishmaniasis. The work is compiled orderly and represented in well-defined manner in the manuscript. The few minor points to consider while final publication are given below.

  1. The abstract is too descriptive. It should be more results oriented.

The abstract was shortened by deleting some more descriptive sentences so that it is now more oriented towards the results.

  1. The authors are advised to add the statistics of disease in the introduction to understand the mortality and severity of the disease.

We agree with the reviewer in highlighting the relevance of the disease. Thus, we have added in the introduction the most recently reported disease burden in DALYs and estimated cases per year.

  1. In manuscript, the final model optimized should be described with parameters obtained from the studies.

We believe that the description of the “best” or “final” model is precise and correct. The model (as all others obtained) is purely based on parameters (i.e. structures and bioactivity data) from the experimental study published earlier. Moreover, all statistical and validation parameters resulting from this model as well as all others are reported in the Supplementary Materials.

  1. The results obtained need to be explained in details with more graphics obtained while study.

We believe that the provided graphics explain the described “best” model very conclusively. The reviewer did not specify which additional graphics she/he would add. In case we receive a request for a specific further figure, we would be happy to provide it.

  1. The manuscript need to be revised for small grammatical mistakes.

We have done our best to use proper English. However, we thank the reviewer for this valuable comment.

We thank the reviewer for the time and effort spent to help us improve our manuscript!

Reviewer 2 Report

My overall impression is good and I find the topic worth investigating. However, I have some suggestions related to this manuscript:

1. Wherever there are MLR models, there are equations which are usually relatively easy to interpret - contrary to many other modelling techniques (including those used in this manuscript), which are perceived as  "black boxes" - you put something in them and you don't  know where the results come from. However, I could not find any MLR equations in the manuscript.

2.   Why didn't the Authors give the results of their predictions for every compound using the particular models?

3. How did the Authors feed the software they used with the compounds'  structures - did they have to draw them manually or could they use e.g. SMILES codes? (If so, please include in Supplementary Materials).

4. There are some details that have to be checked - e.g. Reference 25 - as far as I could see, this paper was published in 2020, not in 2018.

Author Response

Reviewer 2:

  1. Wherever there are MLR models, there are equations which are usually relatively easy to interpret - contrary to many other modelling techniques (including those used in this manuscript), which are perceived as "black boxes" - you put something in them and you don't know where the results come from. However, I could not find any MLR equations in the manuscript.

We thank the reviewer for the suggestion. Although we consider that it is unnecessary to include the equations for those models as their performance was not outstanding, a table (newly generated Table S4) with the corresponding equations was added to the Supplementary Materials (and mentioned in the main text).

  1.   Why didn't the Authors give the results of their predictions for every compound using the particular models?

Due to the number of models evaluated, we decided not to include the predictions for all of them. The respective statistics are already included (using 13 individual parameters and 2 consensus scores). However, we included a new table into the Supplementary Materials (newly generated Table S8) with the actual and predicted activity values (pIC50) obtained by the best model (3D4).

  1. How did the Authors feed the software they used with the compounds' structures - did they have to draw them manually or could they use e.g. SMILES codes? (If so, please include in Supplementary Materials).

All the structures were manually drawn directly into the MOE software.

  1. There are some details that have to be checked - e.g. Reference 25 - as far as I could see, this paper was published in 2020, not in 2018.

We thank the reviewer for the scrutiny and apologize for the mistake. All citations have been checked again for completeness and veracity, modified accordingly in Mendeley, and updated in the manuscript.

We thank the reviewer for the time and effort spent to help us improve our manuscript!
